# Assessment of the Antioxidant and Antimicrobial Potential of *Ptychotis verticillata* Duby Essential Oil from Eastern Morocco: An In Vitro and In Silico Analysis

**DOI:** 10.3390/antibiotics12040655

**Published:** 2023-03-27

**Authors:** Mohamed Taibi, Amine Elbouzidi, Douaae Ou-Yahia, Mohammed Dalli, Reda Bellaouchi, Aziz Tikent, Mohammed Roubi, Nadia Gseyra, Abdeslam Asehraou, Christophe Hano, Mohamed Addi, Bouchra El Guerrouj, Khalid Chaabane

**Affiliations:** 1Laboratoire d’Amélioration des Productions Agricoles, Biotechnologie et Environnement (LAPABE), Faculté des Sciences, Université Mohammed Premier, Oujda 60000, Morocco; 2Centre de l’Oriental des Sciences et Technologies de l’Eau et de l’Environnement (COSTEE), Université Mohammed Premier, Oujda 60000, Morocco; 3Laboratory of Bioresources, Biotechnology, Ethnopharmacology and Health, Faculty of Sciences, Mohammed First University, Boulevard Mohamed VI, B.P. 717, Oujda 60000, Morocco; 4Laboratory of Microbiology, Faculty of Medicine and Pharmacy, University Mohammed The First, Oujda 60000, Morocco; 5Laboratoire de Biologie des Ligneux et des Grandes Cultures, INRAE USC1328, University of Orleans, CEDEX 2, 45067 Orléans, France

**Keywords:** *Ptychotis verticillata* Duby, antioxidant activity, antibacterial activity, antifungal activity, essential oil, computational study, PASS prediction, toxicity in silico, molecular docking

## Abstract

*Ptychotis verticillata* Duby, referred to as Nûnkha in the local language, is a medicinal plant that is native to Morocco. This particular plant is a member of the Apiaceae family and has a longstanding history in traditional medicine and has been utilized for therapeutic purposes by practitioners for generations. The goal of this research is to uncover the phytochemical makeup of the essential oil extracted from *P. verticillata*, which is indigenous to the Touissite region in Eastern Morocco. The extraction of the essential oil of *P. verticillata* (PVEO) was accomplished through the use of hydro-distillation via a Clevenger apparatus. The chemical profile of the essential oil was then determined through analysis utilizing gas chromatography–mass spectrometry (GC/MS). The study findings indicated that the essential oil of *P. verticillata* is composed primarily of Carvacrol (37.05%), D-Limonene (22.97%), γ-Terpinene (15.97%), *m*-Cymene (12.14%) and Thymol (8.49%). The in vitro antioxidant potential of PVEO was evaluated using two methods: the 2,2-diphenyl-1-picrylhydrazyl (DPPH) radical trapping assay and the ferric reducing antioxidant power (FRAP) method. The data demonstrated considerable radical scavenging and relative antioxidative power. *Escherichia coli, Staphylococcus aureus*, *Listeria innocua*, and *Pseudomonas aeruginosa* were the most susceptible bacterial strains tested, while *Geotrichum candidum*, *Candida albicans*, and *Rhodotorula glutinis* were the most resilient fungi strains. PVEO had broad-spectrum antifungal and antibacterial properties. To elucidate the antioxidative and antibacterial characteristics of the identified molecules, we applied the methodology of molecular docking, a computational approach that forecasts the binding of a small molecule to a protein. Additionally, we utilized the Prediction of Activity Spectra for Substances (PASS) algorithm; Absorption, Distribution, Metabolism, and Excretion (ADME); and Pro-Tox II (to predict the toxicity in silico) tests to demonstrate PVEO’s identified compounds’ drug-likeness, pharmacokinetic properties, the anticipated safety features after ingestion, and the potential pharmacological activity. Finally, our findings scientifically confirm the ethnomedicinal usage and usefulness of this plant, which may be a promising source for future pharmaceutical development.

## 1. Introduction

Despite the important progress in medicine during the last few decades, microbial resistance to synthetic and semi-synthetic antimicrobials has been much enhanced, becoming a major health concern in the 21st century [1,2]. As per the World Health Organization (WHO), illnesses that are induced by microorganisms, including bacteria and fungi, are recognized as one of the primary contributors to global morbidity and mortality. Moreover, numerous reports state that microbial resistance will continue to increase, reaching a mortality rate of 10 million in 2050 [3,4]. The scourge of microbial resistance has been ascribed to different factors, primarily the misuse of antimicrobial agents, as well as the widening of the immunosuppressed population, including organ transplant recipients and HIV and cancer patients [5]. The microbial resistance mechanisms to antimicrobials can be intrinsic or acquired. Intrinsic resistance is related to a specific strain and linked to its biological properties. However, acquired resistance is mainly induced by chromosomal mutation, the procurement of the resistance genes by other pathogenic strains, or the combination of both mechanisms [6,7]. Thus, discovering and developing new antimicrobial agents that are effective against resistant microorganisms is an urgent need. Along with finding new antimicrobials, researchers are increasingly looking for potent natural antioxidant molecules to stop the harmful effects of free radicals or reactive oxygen species (ROS). These latter are known to be responsible for many pathologies, such as heart disease and cancer [8]. Medicinal and aromatic flora represent a highly auspicious pool of bioactive compounds, presenting a vast array of molecules with therapeutic potential [9]. For centuries, herbals have been considered as a significant source of antimicrobial compounds and consequently employed in treating infectious diseases [10]. The World Health Organization (WHO) claims that approximately 65–80% of the developing countries’ populations have resorted to medicinal plants for primary healthcare [11]. Moreover, due to the remarkable structural diversity of their compounds, essential oils from many plants have been investigated for their biological properties, especially their antimicrobial and antioxidant activities [12]. Essential oils are organic substances that are extracted naturally from diverse plant parts, including but not limited to leaves, flowers, stems, and roots. These oils are known for their intricate molecular architecture, comprising an amalgamation of various organic compounds. These compounds include terpenoids, which are responsible for their distinctive aromas, ketones, acids, aldehydes, esters, and alcohols [13]. The unique combinations and concentrations of these compounds in essential oils determine their distinct therapeutic properties and effectiveness in combating various ailments. The complex chemical structures of essential oils also confer various modes of action, allowing them to target multiple cellular processes and pathways in microorganisms. This enables essential oils to attack and inactivate microorganisms by disrupting their cellular membranes, inhibiting enzyme activities, and inducing oxidative stress, among other mechanisms [14]. Given their broad-spectrum antimicrobial activity, essential oils are increasingly being studied for their potential use in treating various infectious diseases [14].

*Ptychotis verticillata*, known as Nûnkha and by other several names, including *Psychotis ammoides* Koch., *Petroselium ammoides* Rchb. Fil., *Ammoides pusilla* (Brot.), and Breistr., *Ammoides verticillata* Briq., is a medicinal and aromatic plant widespread in North African countries and widely used in traditional medicine as an antispasmodic, antidiabetic, and antipyretic agent, as well as an antiseptic, and for its antifungal and antibacterial properties [15,16,17]. The medicinal plant *P. verticillata* has been known to have several therapeutic effects and medical benefits, yet, to our understanding, there are very few studies that have explored and reported on its antioxidant and antimicrobial properties in depth. This is an area that deserves more attention. The main objective of this study was to comprehensively examine the chemical composition and antioxidant, antibacterial, and antifungal properties of the major phytocompounds found in PVEO, and to utilize computational techniques (molecular docking) to explore the underlying mechanisms of their interactions.

## 2. Results and Discussion

### 2.1. Composition of the Essential Oil PVEO

The extraction method of hydro-distillation was used to obtain a yield of 5% of PVEO. The chemical composition of PVEO was analyzed and found to consist of 12 compounds, which together constitute 100% of the oil’s total chemical makeup. These compounds are listed in Table 1 and Figure 1, and represented in Figure 2, while MS spectra are provided in Appendix A. The results showed that the oil is primarily composed of phenolic compounds at 45.54%, with particularly high concentrations of Carvacrol (37.05%), and Thymol (8.49%). Other notable constituents include D-Limonene (22.97%), γ-Terpinene (15.97%), and *m*-Cymene (12.14%). These findings differ from a previous study by Tomi et al. (2011), which reported that PVEO is made up of 19 constituents representing 98.9% of the total oil and dominated by high amounts of Carvacrol (44.6%), and Thymol (3.4%) [17]. The research also found that other important constituents of the oil are limonene (18.4%), γ-terpinene (9.5%), *p*-cymene (9.4%), and geranyl acetate (4.7%) [17]. The differences in the oil’s chemical composition may be due to a variety of factors, including biotic and abiotic factors, the age of the plant, the period of its vegetative cycle, or even genetic factors [18,19]. It is worth noting that numerous studies have demonstrated that plants with similar chemical compositions to *P. verticillata* exhibit diverse biological activities, such as antioxidant, antibacterial, antifungal, and insecticidal properties [16,20].

### 2.2. PASS and ADME Prediction

In order to predict the probable antioxidant, antibacterial, and antifungal activities of PVEO compounds, Prediction of Activity Spectra of Substances (PASS), an in silico neural network based-biological activity prediction webserver, was used [21]. This tool is a comprehensive pharmacophore-based virtual screening method that employs a massive library of structures with over 3000 different biological activities. As illustrated in Table 2, the recovered result for each structure was presented as a score known as the Pa score (probable activity) and the Pi score (probable inactivity). Generally, structures with a Pa score more than 0.5 are more likely to exhibit the predicted activity in vitro, whereas those with a Pa score less than 0.5 are unlikely to be active agents [22].

The PASS prediction of the identified phytoconstituents of PVEO revealed that compound **5** (β-myrcene), followed by compounds **11** and **12** (Thymol and Carvacrol, respectively) may be the compounds responsible for the noticed antioxidant activity. For the antibacterial activity prediction, all compounds displayed a modest Pa score, ranging from 0.166 to 0.405. β-myrcene, followed by D-Limonene, were found to have a high Pa score (Pa = 0.582, 0.582, respectively), suggesting that the strong antifungal activity observed may be due to the presence of these two substances. Based on these findings, we projected that PVEO may have a good antifungal and poor antibacterial and antioxidant capabilities, due to the studied components, without overlooking the fact that synergies may be the origin of the powerful antibacterial and antioxidant activity found in the in vitro tests carried out, as described in several published reports [23,24,25,26].

The success of a promising drug can be hindered by its ADMET characteristics, which include its absorption, distribution, metabolism, excretion, and toxicity. The high cost of clinical trials in drug discovery is often due to the drug’s pharmacokinetic properties. To assess the possibility of the PVEO becoming a drug candidate, its ADMET parameters were analyzed using computer-based tools (Table 2, Figure 3). All of the plant constituents were found to conform to Lipinski’s rule of five, and met the criteria set by the Ghose and Veber filters, with most showing good bioavailability (of 0.55) (Figure 4). These compounds were found to share a topological polar surface area (TPSA) less than 30 Å^2^, which suggests good intestinal absorption and good brain penetration. As predicted, all the identified compounds had high Caco-2 permeability values (>0.90) and promising intestinal absorption ranging from 90.84 to 96.30%. Regarding the second pharmacokinetic parameter (distribution), three parameters were taken into consideration. The first one was Log K*p*, indicating skin permeability; it was found that all the compounds had good permeability values, ranging from −4.497 to −2.789 cm/s [27]. VDss, which is expressed as Log VDss (Log L/Kg), refers to the steady-state volume of distribution in human beings. All the compounds exhibited high distribution in plasma, while compounds **5**, **8**, **9**, and **10** were found to have a good distribution. Furthermore, it was observed that all compounds exhibited favorable permeability through the blood–brain barrier (BBB). It is imperative to investigate the metabolic interactions between the compounds and the cytochrome P450 enzyme, as this enzyme plays a crucial role in drug metabolism. This analysis enables us to determine whether the molecule acts as an inhibitor or substrate for the two primary isoenzymes, namely CYP2D6 and CYP3A4. After conducting a thorough examination, it was determined that none of the compounds acted as either a substrate or inhibitor of cytochrome P450 isoforms, as reported in reference [28]. In this study, we have tested in silico the ability of the various phytoconstituents to be transported by Renal Organic Cation Transporter 2 (OCT2). Organic cations are positively charged molecules that are important in drug transport and elimination. Renal OCT2 is responsible for transporting many organic cations, including some drugs, out of the body through the urine [29]. Surprisingly, none of the phytoconstituents were identified as substrates for this transporter, meaning that they were not effectively transported by Renal OCT2. The phytoconstituent β-Myrcene had the best total clearance, with a rate of 0.438 mL/min/kg. This suggests that β-Myrcene is efficiently eliminated from the body through processes other than Renal OCT2 transport.

In this study, we assessed the drug-like characteristics and gastrointestinal absorption of specific compounds derived from PVEO. To accomplish this, we utilized the boiled-egg prediction (as illustrated in Figure 3) and bioavailability radars (displayed in Figure 4). The boiled-egg graph revealed that the compounds located within the yellow yolk zone possessed the ability to permeate through the blood–brain barrier. Furthermore, all compounds analyzed were determined to be non-substrates of P-glycoprotein (PGP-) within this yellow zone. Meanwhile, the bioavailability radars displayed in Figure 4 demonstrate a pink region, which symbolizes the specific domain for oral bioavailability that the molecule must wholly inhabit to be deemed drug-like. Our current investigation uncovered that all phytocompounds conform to the appropriate oral bioavailability area.

### 2.3. Toxicity Prediction Results

Using the webserver Pro-Tox II [30], the toxicity parameters of the selected phytocompounds were predicted. According to the predictions, none of the found phytochemicals, with the exception of compound 7 (*m*-Cymene), which was deemed moderately carcinogenic, were anticipated to be hepatotoxic, carcinogenic, cytotoxic, immunotoxic, or mutagenic. Table 3 also displays the LD_50_ values that were calculated to verify the safety of the identified phytocompounds. The compounds exhibited LD_50_ values exceeding 2000 mg/kg, suggesting their suitability for biological delivery and their potential use as medicinal products.

### 2.4. In Silico Prediction of the Tested Biological Activities

The process of discovering drugs is challenging, and selecting the appropriate lead molecule is critical for the overall success of the project [31,32]. Molecular docking has been the most common computational structure-based drug design (SBDD) approach since the early 1980s [32,33]. The molecular docking methodology is a two-fold process that encompasses two essential phases. Firstly, it involves the anticipation of the shape, orientation, and positioning of the ligand—commonly a minute molecule—within the protein’s binding site, which is referred to as the pose. Secondly, the process demands the assessment of pose excellence, relying on a comprehensive scoring function. It is crucial for the sampling technique to be equipped to replicate the experimental binding mode, while the scoring function must be adept at assigning the highest rank to the top-performing poses amid all generated ones [32].

The current study employed molecular docking to elucidate the potential underlying mechanism responsible for the antioxidant, antifungal, and antibacterial properties of PVEO’s components. The binding affinity values, represented by ΔG, were used to indicate the preference of the compound towards the target in comparison to a known inhibitor. Specifically, this method was utilized to investigate the binding affinities of the 12 compounds present in the essential oil towards four proteins related to bactericidal/bacteriostatic activity, namely DNA Gyrase Topoisomerase II, Enoyl-Acyl Carrier Protein Reductase, Glucosamine-6-Phosphate Synthase, and Penicillin Binding Protein 3 (PDB IDs: 1KZN, 3GNS, 2VF5, and 3VSL, respectively) [34,35,36,37,38,39]. In addition, two proteins related to antifungal activity, Cytochrome P450 14 Alpha-Sterol Demethylase (PDB ID: 1EA1) and N-Myristoyl Transferase (PDB ID: 1IYL), were selected for examination [34,35]. 

The results of molecular docking experiments were presented using a heat map table (Table 4), which utilized a color gradient of red to green to highlight the energies of the docking scores. The lower energy scores were shown in red (usually the same as the native ligand’s docking score), indicating the best matches, while higher energy scores were shown in green. This allowed for the easy identification of chemical compounds that showed the potential inhibition of specific targets.

Topoisomerase enzymes have become a major focus of medication development due to their critical function in DNA duplication [40]. All bacteria include DNA gyrase, a type II topoisomerase that determines the topological condition of the DNA. An essential step in replication or transcription is releasing the supercoiled DNA. As a result, many antibiotics and antimicrobial agents work primarily to halt this phase by blocking DNA gyrase [41]. In light of this, our in silico research has centered around DNA gyrase topoisomerase II (PDB: 1KZN)—an enzyme sourced from the *E. coli* bacterium that is made up of two subunits, GyrA and GyrB. Recent studies have revealed that this enzyme plays a key role in controlling the topological state of bacterial genomes and can be found in all species [42,43]. The compounds detected in PVEO exhibited affinities ranging from −4.6 to −6.2 kcal/mol and did not demonstrate sufficient inhibitory potential. However, the docking of protein 1KZN with its native ligand, Clorobiocin, yielded a strong inhibitory potential with a docking score of −9.6 kcal/mol and the formation of two conventional hydrogen bonds with two amino acid residues (THR A:165, and ASP A:73), as cited in reference [44].

In order for bacteria to build and maintain their cell membranes and other essential cellular structures, they rely on the process of fatty acid biosynthesis. This process involves a series of enzyme-catalyzed reactions that convert simple precursors into the long-chain, unsaturated fatty acids that make up the bulk of these structures. The final step in this pathway is the reduction of the double bond present in an enoyl-ACP derivative, which is an intermediate in the pathway. This step is catalyzed by the enzyme enoyl-acyl carrier protein (ACP) reductase, also known as FabI [45]. The role of FabI is essential for the overall efficiency and effectiveness of the fatty acid biosynthesis pathway. Inhibiting FabI can lead to the accumulation of toxic intermediates and ultimately cell death, making it a promising target for the development of new antibiotics. The main and crucial enoyl reductase enzyme, known as FabI, is extensively present in a variety of microorganisms, including bacterial strains. As in other investigations [46,47], the protein known as enoyl-acyl carrier protein reductase (*S. aureus*) (PDB ID: 3GNS) was selected as a possible target. Scientists are now focusing on a specific enzyme called FabI as a potential target in developing antibacterial compounds. FabI crystal structures have been discovered in various types of bacteria, including *E. coli* and *S. aureus* [48]. Our study has shown that compounds **1**–**12** have moderate to weak binding abilities to FabI when compared to a known ligand called Triclosan (with −6.2 kcal/mol).

Glucosamine-6-Phosphate Synthase (GlmS; PDB ID: 2VF5) is an important rate-limiting enzyme that plays a significant role in the bacterial cell wall. GlmS catalyzes the transformation of fructose-6-phosphate and glutamine to glucosamine-6-phosphate, leading to the end product, which is considered as a building block of the peptidoglycan cell wall, UDP-N-acetylglucosamine. Glucosamine-6-Phosphate inhibits post-transcriptional feedback to the GlmS gene in *E. coli* and other bacteria, being considered as a natural inhibitor in this case. Without G6P synthase, bacteria are unable to survive, which makes it a vital component of antibacterial activity [49,50,51]. When compared to the natural ligand (−7.2 kcal/mol), none of the examined molecules have a high binding affinity toward the targeted protein [52].

Penicillin binding proteins (PBPs) are a group of bacterial enzymes initially named for their interaction with penicillin, but are now recognized as targets for a wider range of antibiotics that possess a β-lactam moiety [53]. PBPs perform an essential function in the creation, maintenance, and modification of the bacterial cell wall’s peptidoglycan structure [54]. Specifically, Penicillin Binding Protein 3 (PDB ID: 3VSL) was identified as the target of the ligands isolated in PVEO. The data obtained from this study showed that the identified ligands exhibited a high binding affinity (ranging from −4.2 to −5.1 kcal/mol) compared to the native ligand, Cefotaxime, which had a binding affinity of −7.1 kcal/mol. The observed antibacterial activity in the in vitro experiments could be attributed to a synergistic mechanism, whereby the essential oils’ volatile, lipophilic compounds interact with the cell wall, enhancing its permeability or altering the cell’s energy generation mechanism, even at low concentrations [55]. Notably, Thymol and Carvacrol have been previously suggested as bioactive agents against *E. coli*, *S. aureus*, and *C. albicans* [56]. Thymol and Carvacrol inhibit the growth of bacteria and their ability to produce ergosterol. They achieve this by disrupting the bacterial membrane, which leads to the leakage of ions and ATP, ultimately resulting in cell death [57,58].

The enzyme Cytochrome P450 14 α-Sterol Demethylase (CYP51s) plays a crucial role in the biosynthesis of sterols in fungi. It catalyzes the production of intermediate compounds in the synthesis of ergosterol. As it is a crucial enzyme in sterol production [59], CYP51s has been identified as a target for antifungal drugs [60]. One such study found that the compound (+)-4-Carene has a strong inhibitory effect on the fungal protein, with a binding energy of −6.1 kcal/mol, which is more potent than the existing antifungal drug Fluconazole (−5.8 kcal/mol). This compound interacts with specific amino acid residues in the enzyme’s active site by forming seven hydrogen bonds with ILE A:27, ARG A:274, ARG A:354, TYR A:426, ARG A:427, ASN A:428, and HIS A:430 (Appendix A). These amino acids were found to play a critical role in forming the active site pocket of the protein [34].

N-Myristoyl Transferase (NMT) is a protein that is exclusively present in eukaryotic cells. Its primary function is to catalyze the binding of myristate fatty acid to the N-terminal glycine of other proteins, utilizing myristoyl-CoA as a substrate [61]. NMT is an essential component in a variety of biological processes, such as signal transduction, cell death, and the growth of fungal pathogens [62]. For this study, the researchers selected N-Myristoyl Transferase (PDB ID: 1IYL), which suggests that the observed antifungal activity is not a result of inhibiting this protein.

Lipoxygenases are a group of enzymes that utilize a redox mechanism to catalyze the oxidation of polyunsaturated fatty acids. This reaction produces an oxygen-centered radical from the fatty acid, called a hydroperoxide. The formation of such a radical can contribute to the development of various severe diseases [63,64]. Two proteins, lipoxygenase (1N8Q) and cytochrome P450 (1OG5), were chosen for study. For lipoxygenase, five ligands were identified as strong inhibitors, α-Thujene, (+)-4-Carene, D-Limonene, Carvacrol, and Thymol, with binding affinity values of −6.5, −6.1, −6.0, −6.2, and −6.2 kcal/mol, respectively, relative to the native ligand (Protocatechuic Acid), which had a binding energy value of −6.0 kcal/mol. Of these inhibitors, α-Thujene was found to have the lowest binding affinity, making it the most potent inhibitor among all those tested and forming five hydrogen bonds with the amino acids GLN A:514, HIS A:518, ILE A:557, LEU A:773, and ILE A:857 (Appendix A). However, none of the ligands for the second targeted antioxidant protein, Cytochrome P450 (PDB ID: 1OG5), showed a free binding energy lower than −6.6 kcal/mol (that of the native inhibitor, Warfarin [65]), indicating that PVEOs had a minimal capacity to inhibit Cytochrome P450. Regarding the latter protein, none of the docked molecules are NADPH oxidase protein inhibitors (PDB ID: 2CDU).

PVEO’s plant-based molecules have exhibited the potential to hinder the function of bovine serum albumin (BSA) protein, evidenced by their interaction with the BSA protein (PDB: 4JK4) (Appendix A). Moreover, among the inhibitors, *m*-Cymene, Carvacrol, and β-Myrcene were found to be the most effective against the BSA protein, with binding affinities of −7.4, −6.4, and −6.4 kcal/mol, respectively, when compared to the native ligand, 3,5-Diiodosalicylic Acid, with a binding affinity of −5.3 kcal/mol. It is speculated that the antioxidant properties of the native ligands are linked to the particular amino acid residues to which they bind. This assumption is based on the common amino acid residues, binding strengths, available literature, and the presence of natural inhibitors [66]. The results indicate that PVEO has a promising antioxidant effect.

### 2.5. Antioxidant Activity

In order to precisely assess the antioxidant potential of PVEO, two methods were utilized, namely DPPH and iron-reducing power (FRAP). The IC_50_ value, which denotes the quantity of antioxidants needed to decrease the concentration of free radicals by 50%, is inversely correlated with the antioxidant activity.

Upon analyzing the results obtained, it was found that PVEO exhibited a noteworthy level of antioxidant activity, with IC_50_ values of 70.6 ± 0.005 μg/mL and 50 ± 0.001 μg/mL, for the DPPH and FRAP assays, respectively (Table 5). However, these antioxidant potentials were lower than those found for the ascorbic acid used as a control, with IC_50_ values ranging from 21.06 ± 0.001 µg/mL to 19.33 ± 0.001 µg/mL. Our findings are in accordance with Tomi et al. (2011), who evaluated the antioxidant activity of PVEO, using the DPPH assay, and reported that the EO exhibited important scavenging potency, suggesting it as a source of potential natural antioxidants [17].

The antioxidant capacity observed in our analyzed EO is most likely attributed to its chemical makeup, with phenolic compounds being the dominant component. Phenolic compounds are considered highly effective natural antioxidants due to their chemical structure, which enables them to scavenge free radicals and convert them into stable compounds through proton and electron transfer mechanisms [67,68]. It is worth noting that the antioxidant potential of EOs cannot be solely attributed to their major constituents, as even minor components can contribute to their antioxidative potential through synergistic effects [69].

### 2.6. Antibacterial Activity

#### Minimum Inhibitory Concentration (MIC) and Bactericidal Concentration (MBC)

The inhibition zone diameters were assessed by the disc diffusion assay, and the MIC and MBC of the essential oil studied were evaluated using the microdilution method (Table 6 and Figure 5). Essential oils are considered as active compounds if the diameters of their microbial growth inhibition are greater than or equal to 15 mm [70]. The essential oil was found to have strong antibacterial properties against all of the bacterial species that were tested, as shown by the results. The growth inhibition zone diameters varied between 23 and 46 mm. The highest inhibition zone diameter (IZ = 46.25 mm) was observed against *Escherichia coli*, while the lowest inhibition zone diameter (IZ = 22.5 mm) was observed against *Pseudomonas aeruginosa*. According to the results presented in Table 6, the studied EO exerted significant inhibitory activity against all the studied bacteria: Gram-positive (*L. innocua* and *S. aureus*) and Gram-negative (*P. aeruginosa* and *E. coli*). In particular, *E. coli* and *S. aureus* showed high sensitivity to this oil, even at low concentrations of 0.25%. Furthermore, the study found that a concentration of only 0.5% of the EO was sufficient to inhibit the growth of both *L. innocua* and *P. aeruginosa*, two of the bacterial species tested. This suggests that even a relatively low concentration of the essential oil may be effective at preventing the growth and spread of these bacteria.

The chemical composition results showed that PVEO is dominated by Thymol and Carvacrol; the interactions between these constituents may also affect their activity. A study conducted by Lambert et al. in 2001 examined the antibacterial properties of an essential oil against *S. aureus* and *P. aeruginosa*. The investigation revealed that the EO was successful in suppressing the growth of the two bacterial strains. However, the precise mechanism behind its action remains unclear [57]. The research team isolated two major components of the essential oil, Carvacrol and Thymol, and tested their individual effects against the bacteria. They found that both Carvacrol and Thymol had some antibacterial activity, but when used together, they exhibited a synergistic effect. This means that the combination of the two compounds was more effective at inhibiting bacterial growth than either compound alone. Another synergistic interaction has been demonstrated between Carvacrol and *m*-Cymene, on Gram-positive bacteria. *m*-Cymene seemed to facilitate the intracellular penetration of Carvacrol, thus potentiating its action [71]. Thus, the chemical composition of PVEO and the synergy between these components may represent the leading cause of the active antibacterial principle. The results shown in Table 3 show that the studied EO has bactericidal activity towards both types of Gram-negative and Gram-positive strains, with an MBC of 1% for all the studied strains.

The studied EO showed potential antibacterial activity against all tested strains. The bacterial strains showed almost the same sensitivity to the studied essential oil despite the difference in Gram. These results do not agree with those stated by Gachkar et al. (2007), who mentioned that there is a difference in the inhibitory activity observed between Gram-positive and Gram-negative bacteria due to the composition of their cell walls. For Gram-positive bacteria, the inhibitory activity is based mainly on the direct contact of their hydrophobic compounds with the phospholipids of the cell membrane, which could induce structural damage or the total rupture of the cell membrane, a loss of cell constituents, and internal control of the cell. However, the resistance of Gram-negative cells is attributed to diffusion constraints across their outer membranes, caused by a hydrophilic barrier [72].

### 2.7. Antifungal Activity

#### Minimum Inhibitory Concentration (MIC) and Fungicide (MFC)

To assess the degree of inhibition zone, the disc diffusion assay was employed, while the microdilution technique was utilized for calculating the minimum inhibitory concentration (MIC) and minimum fungicidal concentration (MFC) of the essential oil in question. EO components are regarded as active compounds if, as previously mentioned, the microbial growth inhibition diameters are more than 15 mm [70]. PVEO showed moderated antifungal action on the tested fungi species. The diameters of the growth inhibition zone varied between 19.5 and 31.75 mm (Figure 6). The inhibition value against the two yeasts (*R. glutinis* and *C. albicans*) was lower than that against the mold (*G. candidum*). Due to their chemical composition, PVEO has growth-inhibiting properties against many yeasts and molds. Numerous in vitro investigations have examined these compounds’ antifungal abilities [73].

Both the MFC and MIC of the investigated essential oil were assessed (Table 7). According to Table 7’s antifungal activity data, PVEO had an identical minimum inhibitory concentration value of 1% for all three strains of *C. albicans*, *R. glutinis*, and *G. candidum* that were examined. The essential oil under examination exhibited efficacy in inhibiting the growth of various types of fungi. This activity depends on the factors that react together to obtain an antifungal action. The composition of this EO may be one of the most critical parameters influencing the antifungal activity. The elevated capacity of the substance can be primarily attributed to the presence of terpene compounds, which are accountable for the antifungal properties through their actions of membrane disturbance, breakdown of the fungal mitochondria, and inhibition of electron transport and mitochondrial ATPase. Furthermore, phenolic compounds are also responsible for cell membrane impairment [74].

## 3. Materials and Methods

### 3.1. Plant Material

*Ptychotis verticillate*, a plant that is endemic to the north-eastern region of Morocco, was the subject of this study. The aerial part of the plant was first collected from a local market (the plant was harvested in Spring 2022) in the vicinity of Oujda, Morocco, and brought to the Faculty of Sciences at the University Mohammed the First in Oujda, for taxonomical identification. The specimen, which was in good condition, was subsequently preserved in the herbarium of the Faculty of Sciences, as a voucher specimen, and it was assigned the number HUMPOM17. This step is crucial for future reference, as it allows for the precise identification of the plant species, which is essential for botanical research and the management of Morocco’s biodiversity.

### 3.2. Essential Oil Extraction

To extract the EO, the hydrodistillation method was used and this process was performed using a modified Clevenger setup. The procedure consisted of introducing 100 g of the plant’s aerial part into a 5-L flask containing 400 mL of distilled water. Once the plant material and water were loaded, the apparatus was closed and set up properly. Then, the heating source was turned on and adjusted to the optimum temperature for stable and consistent extraction, which was set to 100 °C. The constant and controlled heating condition is a vital step to ensure that the extraction process is efficient and yields a high-quality EO.

### 3.3. Qualitative and Semi-Quantitative Analysis of the PVEO by GC/MS

The qualitative and semi-quantitative analysis of PVEO was conducted using a gas chromatograph with a mass spectrometer detector, as previously described in [46]. A Shimadzu GC system from Kyoto, Japan was utilized in combination with an MS QP2010 (Shimadzu Scientific Instruments, Kyoto, Japan) to identify and separate compounds. Separation was achieved using a BPX25 capillary column with a 95 percent dimethylpolysiloxane diphenyl phase with a 30 m length, 0.25 mm internal diameter, and 0.25 m film thickness. Pure helium (99.99 percent) was used as the carrier gas, at a constant flow rate of 3 mL/min, and the injection, ion source, and interface temperatures were fixed at 250 °C. The column furnace was programmed to increase from 50 °C (for 1 min) to 250 °C at a rate of 10 °C/min, maintaining the temperature for 1 min. Sample components were ionized in EI mode at 70 eV, and the mass range studied was 40 to 300 *m*/*z*. Each produced oil was fed into the chamber at 1 L, diluted with an appropriate solvent. Then, 1 μL of each prepared oil was injected in fractionation mode (fractionation ratio 90:1). Three evaluations were conducted per sample, and compounds were identified by comparing the retention times to verified standards and mass spectrum fragmentation models found in databases or on NIST compounds. Laboratory Solutions (v2.5) was used to collect and process data.

### 3.4. PASS, ADME, the Prediction of the Toxicity Analysis (Pro-Tox II)

To predict the pharmacological activity of the main chemical constituents of PVEO, the researchers utilized the PASS method, a computational tool used for predicting the biological activity of chemical compounds. It uses a statistical algorithm to compare the chemical structure of a compound with a large database of known bioactive compounds, and predicts the likelihood of the compound exhibiting certain activities, such as binding to specific receptors, inhibiting enzymes, or affecting metabolic pathways [75]. First, the chemical compounds were transformed into SMILES format through ChemDraw and then analyzed using the PASS online program, which indicated the probable activity (Pa) and likely inactivity (Pi) of the drug-like compounds [21,76]. Absorption, Distribution, Metabolism, and Excretion are key factors that determine the pharmacokinetic profile of a compound, or how it is absorbed, distributed, metabolized, and eliminated from the body. Computational tools are used to predict the ADME properties of compounds, such as their permeability across cell membranes, their affinity for transporters and enzymes involved in drug absorption and elimination, and their metabolic stability. For this reason, we assessed the physicochemical properties, drug similarity, and pharmacokinetic properties of the compounds using ADME webservers such as SwissADME (http://www.swissadme.ch/ accessed on 5 February 2023) and pkCSM (http://biosig.unimelb.edu.au/pkcsm/ accessed on 5 February 2023) [28,77,78].

To estimate toxicity levels, the Pro-Tox II online tool (https://tox-new.charite.de/protox_II/ accessed on 5 February 2023) was used. This online tool uses a statistical algorithm to compare the chemical structure of a compound with a large database of known toxic compounds, and predicts the likelihood of the compound causing toxicity or adverse effects in humans or other organisms. This tool provided information on the LD_50_ values, toxicity class, and the toxicological endpoints, including hepatotoxicity, carcinogenicity, immunotoxicity, mutagenicity, and cytotoxicity [30]. These methods and tools provided valuable insights into the potential therapeutic applications and toxicity risks of PVEO compounds.

### 3.5. Molecular Docking Protocol

To investigate the potential medicinal properties of PVEO, we used molecular docking to predict the antioxidant, antibacterial, and antifungal activity of the molecules identified in the essential oil. The procedure followed the well-established protocol detailed in previous studies [46,64]. To perform the molecular docking study, the three-dimensional (3D) structures of the phytocompounds present in PVEO were obtained from PubChem (https://pubchem.ncbi.nlm.nih.gov/ accessed on 1 December 2022). These structures were downloaded in “3D sdf” format and were then transformed into “pdb” format using PyMol, a software program for visualizing and analyzing molecular structures. To examine how the phytocompounds in PVEO interacted with target proteins, the crystallographic structures of the proteins were obtained from the Protein Data Bank website (https://www.rcsb.org/, accessed on 1 December 2022), using their specific PDB IDs. The structures were visualized using the Discovery Studio 4.1 software, which enables the detailed examination of protein structures [79]. Before performing the molecular docking study, it was necessary to prepare the protein structures by removing the typical inhibitors, water molecules, and ions present in them. This process was done using a well-established method. Moreover, to make the model more accurate, polar hydrogen bonds and Kollmann charges were added to the protein structures, which helped to consider the electrostatic interactions during the docking process [79]. The automated docking experiments were performed using the AutoDock Vina v1.5.6 software [80]. The process of converting the protein and ligands into three-dimensional PDBQT files was done by utilizing MGL tools. The software AutoGrid, a component of AutoDock, was used to construct grid maps that showed the energy of interaction between the macromolecule target and the ligands during the docking experiment.

In order to improve the accuracy of the docking process, the search space grid box was enlarged, as indicated in Table 8. The binding energies (∆G) of the ligand complexes were calculated and reported in Kcal/mol. With the aid of the Discovery Studio 4.1 software, 2D diagrams illustrating molecular interactions were generated and analyzed to assess the binding interactions between the proteins and ligands. Furthermore, relevant target proteins were identified from the existing literature to explore the possible mechanisms of action of the compounds discovered in PVEO. The study focused on four antibacterial proteins, namely DNA Gyrase Topoisomerase II (PDB ID: 1KZN), Enoyl-Acyl Carrier Protein Reductase (PDB ID: 3GNS), Glucosamine-6-Phosphate Synthase (PDB ID: 2VF5), and Penicillin Binding Protein 3 (PDB ID: 3VSL) [35,36,37,38,39], as well as two antifungal target proteins, namely Cytochrome P450 14 α-Sterol Demethylase (PDB ID: 1EA1) and N-Myristoyl Transferase (PDB ID: 1IYL) [34,35]. Moreover, the study selected four protein structures as antioxidant proteins, namely lipoxygenase, CYP2C9, NADPH oxidase, and bovine serum albumin (PDB IDs: 1N8Q, 1OG5, 2CDU, and 4JK4, respectively) [65,81].

### 3.6. Antioxidant Activity

#### 3.6.1. 2,2-Diphenyl-1-Picrylhydrazyl (DPPH) Scavenging Assay

The effectiveness of antioxidant activity was determined through the application of the 2,2-diphenyl-1-picrylhydrazyl (DPPH) scavenging assay, which was utilized to establish the antioxidant properties of the substance in question, as reported in [82], with slight modification. The essential oil was diluted in series and 1 mL of each dilution was combined with 1 mL of 0.1 mM methanolic DPPH solution. The mixture was incubated in the dark for 30 min, after which the optical density was measured at 517 nm. The scavenging activity was then calculated using the provided formula:$$\mathrm{Radical}\;\mathrm{Scavenging}\;\mathrm{Activity}\;(\%)=(\frac{A_{0}-A_{1}}{A_{0}})\;\times \;100$$
where A_0_ is the absorbance in the absence of the sample; and A_1_ is the absorbance in the presence of the sample. Ascorbic acid served as the reference. Each experiment was performed three times.

#### 3.6.2. Ferric-Reducing Antioxidant Power (FRAP)

The ferric-reducing antioxidant power (FRAP) method, as described by Ibrahimi and colleagues in 2020 [83], was employed to assess the ferric-reducing antioxidant activity of the extract. To accomplish this, the extract was subjected to various doses ranging from 0.1 to 0.8 mg/mL. One milliliter of each dose was then introduced to a mixture comprising 2.5 mL of 0.2 M phosphate buffer (pH = 6.6) and 2.5 mL of a 1% K_3_[Fe(CN)_6_] solution. The resulting mixture was incubated at 50 °C for 20 min and subsequently cooled to room temperature. The reaction was halted by the addition of 2.5 mL of trichloroacetic acid (10%) and centrifugation of the mixture at 3000 rpm for 10 min. The resulting supernatant (2.5 mL) was combined with 2.5 mL of distilled water and 0.5 mL of 0.1% FeCl_3_, 6 H_2_O solution. Absorbance was measured at 700 nm, and ascorbic acid was utilized as the standard. All tests were executed thrice.

### 3.7. Antimicrobial Activity

#### 3.7.1. Bacterial Strains and Growth Conditions

The effectiveness of the essential oil as an inhibitor of bacterial growth was determined through testing on four pure strains sourced from the Laboratory of Microbial Biotechnology at the Faculty of Science in Oujda, Morocco. These strains included two Gram-positive bacteria (*Staphylococcus aureus* (ATCC 6538) and *Listeria innocua* (ATCC 49.189)) and two Gram-negative bacteria (*Escherichia coli* (ATCC 10536) and *Pseudomonas aeruginosa* (ATCC 15442)). The bacteria were grown on Luria–Bertani agar medium and incubated at 37 °C for 24 h. The concentration of bacteria was measured and adjusted to 106 cells/mL using a UV–visible spectrophotometer at 620 nm before the essential oil was tested.

#### 3.7.2. Disc Diffusion Method

The initial screening for the antimicrobial activity against mycobacteria was done by applying the disc diffusion method, which is a widely used method for testing antimicrobial susceptibility. This method was utilized in compliance with guidelines established by the National Committee for Clinical Laboratory Standards. This approach allows for the efficient evaluation of the substance’s ability to inhibit the growth of mycobacteria [84]. The procedure involved adding a prepared mycobacterial inoculum to Petri dishes containing Sauton agar growth media, on which filter paper discs with a 6 mm diameter were placed, impregnated with 5 mL of each essential oil. These plates were incubated at 37 °C for bacterial strains and 25 °C for fungal strains for a period of 48–72 h, after which they were stored at 4 °C for 2 h to improve molecular diffusion. The size of the inhibitory zone surrounding the disc was measured as an indicator of the antimycobacterial activity. Each experiment was repeated 3 times for accuracy.

#### 3.7.3. Determination of the MIC and the MBC

To produce an emulsion of essential oils as explicated in reference [85], Mueller–Hinton culture medium containing 0.15% agar was utilized, given the insolubility of essential oils in both water and culture media. The minimum inhibitory concentration (MIC) was assessed across a range of essential oil concentrations from 4 to 0.0078% utilizing 96-well microplates and the microdilution method, as delineated in reference [86]. Bacterial inoculum adjusted to 106 cells/mL was added to each well, and gentamicin was implemented as a positive control. The microplates were then subjected to incubation at 37 °C for 24 h. To verify the existence of bacteria, 15 µL of 0.015% resazurin was introduced to the wells and they were further incubated for 2 h at 37 °C, with observations made for the transformation of blue-colored resazurin to pink-colored resorufin, as depicted in reference [87]. Each assessment was performed in triplicate.

To ascertain the minimum bacterial concentration (MBC), a 3 µL sample from the negative control wells was transferred to a nutrient-rich growth medium called MHA and incubated at 37 °C for 24 h. After the incubation period, the MBC was established as the minimum concentration of the essential oil that resulted in no observable bacterial growth.

### 3.8. Antifungal Activity

#### 3.8.1. Choice and Origin of Strains

The antifungal activity of the studied PVEO was tested on three pure fungal strains, *Geotrichum candidum*, *Candida albicans*, and *Rhodotorula glutinis* (ON 209167), from the Microbial Biotechnology Laboratory of the Faculty of Sciences, Oujda, Morocco.

#### 3.8.2. Preparation of the Inoculum and Disc Diffusion Method

The *G. candidum* strain was cultivated on a Potato Dextrose Agar (PDA) medium supplied by BIOKAR in France, at 25 °C for a period of seven days. After this time, the spores’ concentration was precisely adjusted to 2 × 10^6^ spores/mL using a hemacytometer (Thoma cell). Similarly, *C. albicans* and *R. glutinis* were grown on Yeast Extract Peptone Dextrose (YPD) medium at 25 °C for 48 h. The concentration of cells was then adjusted to 10^6^ cells/mL for each yeast strain. As mentioned in Section 3.7.2, the disc diffusion technique was used for the main antimycobacterial screening, as advised by the National Committee for Clinical Laboratory Standards [84].

#### 3.8.3. Determination of the MIC and the MFC

The present study employed the 96-well microplate method to determine the minimum inhibitory concentration (MIC) for the strains of *C. albicans* and *R. glutinis*, using a concentration range of 8% to 0.0015%. Following the incubation of the microplates at 25 °C for 48 h, 15 microliters of resazurin solution were added to each well to monitor growth, with a subsequent 2 h incubation period at 25 °C. The MIC was then determined as the lowest concentration of essential oils resulting in a color change in the resazurin solution from blue to pink, signifying growth. For *G. candidum*, PVEO dilution was carried out by PCB with 0.15% agar. The technique used was the 96-well microplate technique with a concentration range from 8% to 0.0015%. The concentration of the fungal suspension was in the order of 2 × 10^8^ spores/mL, and incubation was carried out at 25 °C for 72 h. A positive control, Cycloheximide, was also used. After the incubation period, the presence of fungal growth was detected using the resazurin method.

The minimum fungicidal concentration (MFC) was determined by transferring a small quantity (3 µL) of the contents from wells that exhibited no growth onto YEG medium, which was then incubated for 48 h at 25 °C for *C. albicans*, and for 72 h at 25 °C for *R. glutinis* and *G. candidum*, respectively.

## 4. Conclusions

The present investigation has revealed that the EO obtained from *P. verticillata* is rich in monoterpenes, such as α-Thujene, α-Pinene, Sabinene, β-Pinene, β-Myrcene, and (+)-4-Carene, among others. Moreover, the essential oil has been shown to contain significant concentrations of Thymol and Carvacrol. Online theoretical studies using PASS prediction have predicted modest antioxidant, antifungal, and antibacterial properties. The ADME analysis revealed that PVEO has drug-like properties, meeting Lipinski’s rule of five, and the Ghose and Veber filters, and having a high gastrointestinal absorption value while having a low impact on CYP450 enzymes. The prediction of the toxicological aspects of the studied essential oil indicated its safety and suitability for biological delivery, and as a prospective medicine. The docking results of the identified compounds to their receptors showed that they are predicted to have high inhibition potential towards the antioxidant receptors (1N8Q and 4JK4), low inhibitory potential towards the studied antibacterial targets, and a high affinity with the antifungal receptor (1EA1). The experimental results demonstrated the unexpected potential of PVEO in exhibiting a potent antibacterial effect on two Gram-negative bacteria (*E. coli* and *P. aeruginosa*) and two Gram-positive strains (*L. innocua* and *S. aureus*), and good antifungal potency against *R. glutinis*, *C. albicans*, and *G. candidum.* PVEO demonstrated potent antioxidant activity, which was demonstrated using DPPH and FRAP assays. Thus, it can be inferred that the PVEO examined in this study holds promise as a natural source of antioxidants and preservatives for use in food and therapeutic products.

## Figures and Tables

**Figure 1 antibiotics-12-00655-f001:**
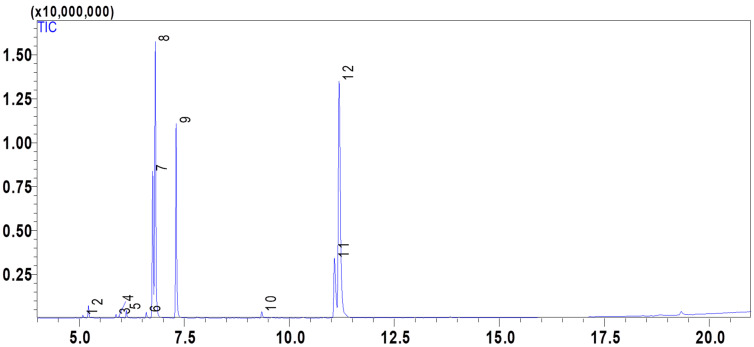
GC/MS chromatogram of the chemical composition of *P. verticillata* essential oil, PVEO. (**1**) α-Thujene, (**2**) α-Pinene, (**3**) Sabinene, (**4**) β-Pinene, (**5**) β-Myrcene, (**6**) (+)-4-Carene, (**7**) *m*-Cymene, (**8**) D-Limonene, (**9**) γ-Terpinene, (**10**) *p*-Menth-1-en-4-ol, (**11**) Thymol, (**12**) Carvacrol.

**Figure 2 antibiotics-12-00655-f002:**
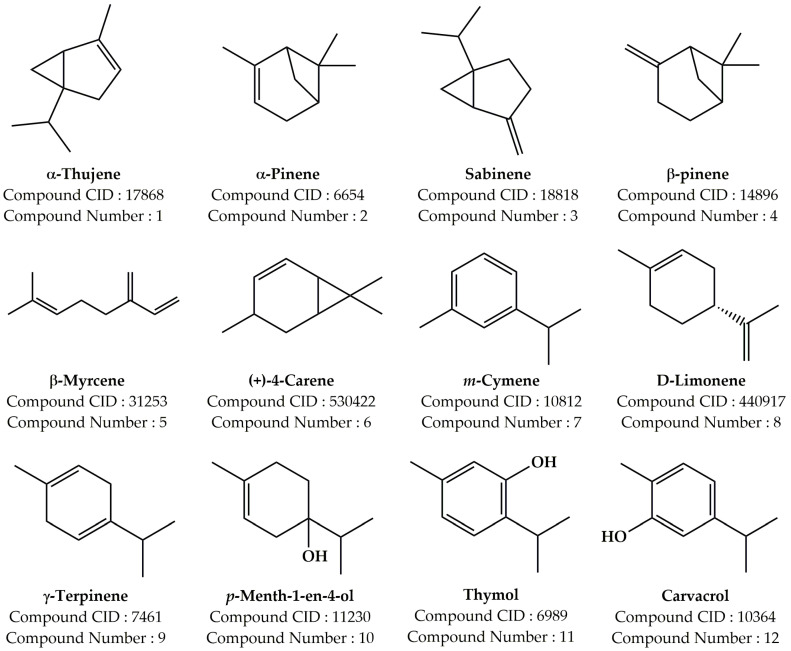
The structures of the chemical constituents found in the essential oil of *P. verticillata* (PVEO).

**Figure 3 antibiotics-12-00655-f003:**
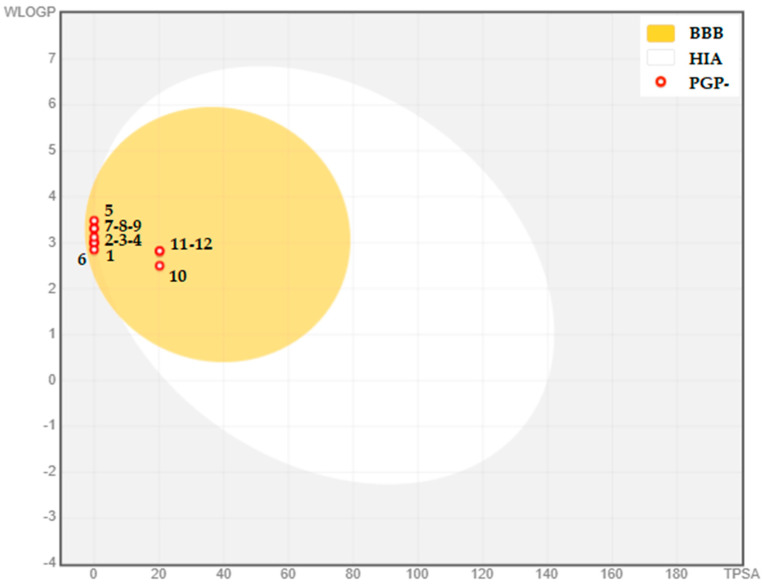
Boiled-egg scheme of the identified components in PVEO. (**1**) α-Thujene, (**2**) α-Pinene, (**3**) Sabinene, (**4**) β-Pinene, (**5**) β-Myrcene, (**6**) (+)-4-Carene, (**7**) *m*-Cymene, (**8**) D-Limonene, (**9**) γ-Terpinene, (**10**) *p*-Menth-1-en-4-ol, (**11**) Thymol, (**12**) Carvacrol.

**Figure 4 antibiotics-12-00655-f004:**
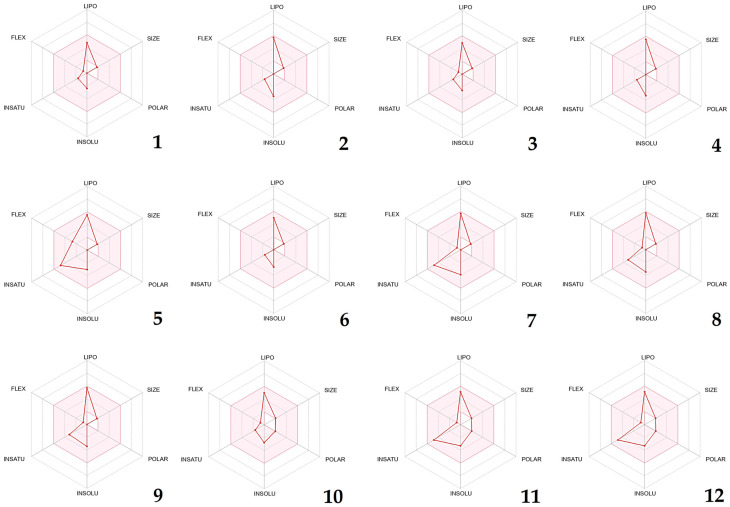
Bioavailability radars of PVEO’s compounds. (**1**) α-Thujene, (**2**) α-Pinene, (**3**) Sabinene, (**4**) β-Pinene, (**5**) β-Myrcene, (**6**) (+)-4-Carene, (**7**) *m*-Cymene, (**8**) D-Limonene, (**9**) γ-Terpinene, (**10**) *p*-Menth-1-en-4-ol, (**11**) Thymol, (**12**) Carvacrol.

**Figure 5 antibiotics-12-00655-f005:**
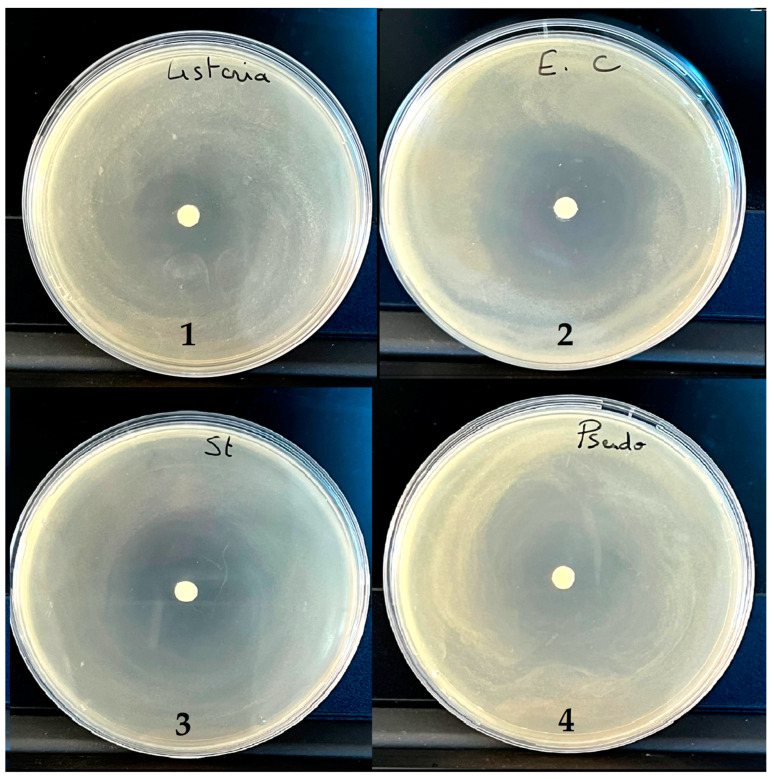
Images of inhibition zones of *Listeria inocua* (**1**), *Escherichia coli* (**2**), *Staphylococcus aureus* (**3**), and *Pseudomonas aeruginosa* (**4**) treated with PVEO.

**Figure 6 antibiotics-12-00655-f006:**
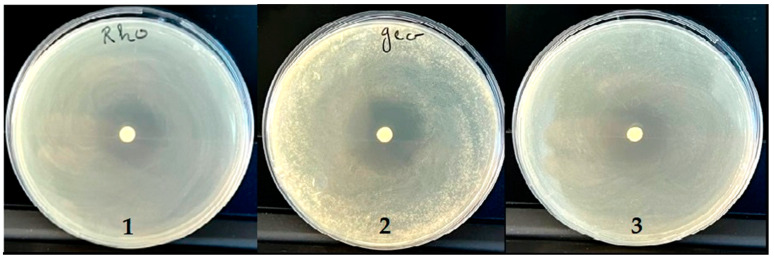
Images of inhibition zones of *Rhodotrula glutinis* (**1**), *Geotrichum candidum* (**2**), and *Candida albicans* (**3**), treated with PVEO.

**Table 1 antibiotics-12-00655-t001:** GC-MS analysis of the phytochemical composition of *P. verticillata* essential oil (PVEO).

Compound Number	Compound Name	Formula	Mol. Wt.	RT (min)	Peak Area (%)
**1**	α-Thujene	C_10_H_16_	136.23	5.086	0.21
**2**	α-Pinene	C_10_H_16_	136.23	5.216	0.95
**3**	Sabinene	C_10_H_16_	136.23	5.872	0.29
**4**	β-Pinene	C_10_H_16_	136.23	5.945	0.16
**5**	β-Myrcene	C_10_H_16_	136.23	6.118	0.62
**6**	(+)-4-Carene	C_10_H_16_	136.23	6.593	0.48
**7**	*m*-Cymene	C_10_H_14_	134.22	6.745	12.14
**8**	D-Limonene	C_10_H_16_	136.23	6.806	22.97
**9**	γ-Terpinene	C_10_H_16_	136.23	7.302	15.97
**10**	*p*-Menth-1-en-4-ol	C_10_H_18_O	154.25	9.342	0.67
**11**	Thymol	C_10_H_14_O	150.22	11.072	8.49
**12**	**Carvacrol**	**C_10_H_14_O**	**150.22**	**11.183**	**37.05**

**Table 2 antibiotics-12-00655-t002:** In silico Prediction of Spectral Activity of Substances (PASS), drug-likeness, and the pharmacokinetic properties (ADME) of the identified components in PVEO. (**1**) α-Thujene, (**2**) α-Pinene, (**3**) Sabinene, (**4**) β-Pinene, (**5**) β-Myrcene, (**6**) (+)-4-Carene, (**7**) *m*-Cymene, (**8**) D-Limonene, (**9**) γ-Terpinene, (**10**) *p*-Menth-1-en-4-ol, (**11**) Thymol, (**12**) Carvacrol.

Prediction	1	2	3	4	5	6	7	8	9	10	11	12
	PASS Prediction (Pa/Pi)											
Antioxidant	n.d.	n.d.	n.d.	n.d.	0.470/0.008	n.d.	0.140/0.115	0.157/0.094	n.d.	0.151/0.102	0.299/0.023	0.302/0.023
Antibacterial	0.166/0.149	0.326/0.051	0.201/0.117	0.233/0.093	0.398/0.030	0.378/0.036	0.213/0.106	0.405/0.029	0.325/0.051	0.328/0.050	0.336/0.047	0.319/0.053
Antifungal	0.337/0.067	0.439/0.042	0.340/0.066	0.225/0.121	0.584/0.020	0.448/0.040	0.343/0.065	0.582/0.020	0.443/0.041	0.466/0.036	0.464/0.037	0.449/0.039
	**Drug-Likeness Prediction**											
**Lipinski**	Yes	Yes	Yes	Yes	Yes	Yes	Yes	Yes	Yes	Yes	Yes	Yes
**Egan**	Yes	Yes	Yes	Yes	Yes	Yes	Yes	Yes	Yes	Yes	Yes	Yes
**Veber**	Yes	Yes	Yes	Yes	Yes	Yes	Yes	Yes	Yes	Yes	Yes	Yes
**Bioavailability score**	0.55	0.55	0.55	0.55	0.55	0.55	0.55	0.55	0.55	0.55	0.55	0.55
	**ADME Prediction** **Physicochemical Properties**											
**TPSA (Å^2^)**	0.00 Å²	0.00 Å²	0.00 Å²	0.00 Å²	0.00 Å²	0.00 Å²	0.00 Å²	0.00 Å²	0.00 Å²	20.23 Å²	20.23 Å²	20.23 Å²
	**Absorption Parameters**											
**Water Solubility**	−4.294	−3.733	−4.629	−4.191	−4.497	−3.492	−4.098	−3.568	−3.941	−2.296	−2.789	−2.789
**Caco-2 Permeability**	1.386	1.38	1.404	1.385	1.400	1.392	1.526	1.401	1.414	1.502	1.606	1.606
**Intestinal Absorption (%)**	95.25	96.04	95.35	95.52	94.69	96.30	93.64	95.89	96.21	94.01	90.84	90.84
**Solubility Class**	Soluble	Soluble	Soluble	Soluble	Soluble	Soluble	Soluble	Soluble	Soluble	Soluble	Soluble	Soluble
	**Distribution Parameters**											
**Log K*_p_* (cm/s)**	−5.11	−3.95	−4.94	−4.18	−4.17	−4.82	−3.92	−3.89	−3.94	−4.93	−4.87	−4.74
**VDss**	0.575	0.667	0.566	0.685	0.363	0.514	0.724	0.396	0.412	0.210	0.512	0.512
**BBB Permeability**	Yes	Yes	Yes	Yes	Yes	Yes	Yes	Yes	Yes	Yes	Yes	Yes
	**Metabolism Parameters**											
**CYP2D6 and CYP3A4 Substrate**	No	No	No	No	No	No	No	No	No	No	No	No
**CYP2D6 and CYP3A4 Inhibitors**	No	No	No	No	No	No	No	No	No	No	No	No
	**Excretion Parameters**											
**Total Clearance**	0.077	0.043	0.071	0.030	0.438	0.029	0.249	0.213	0.217	1.269	0.211	0.207
**Renal OCT2 Substrate**	No	No	No	No	No	No	No	No	No	No	No	No

n.d.: not determined; TPSA: Topological Polar Surface Area; Water Solubility: log mol/L; VDss: log L/kg; Total Clearance: log (mL/min/kg).

**Table 3 antibiotics-12-00655-t003:** Assessment of the toxicological properties of PVEO’s compounds using Pro-Tox II. (**1**) α-Thujene, (**2**) α-Pinene, (**3**) Sabinene, (**4**) β-Pinene, (**5**) β-Myrcene, (**6**) (+)-4-Carene, (**7**) *m*-Cymene, (**8**) D-Limonene, (**9**) γ-Terpinene, (**10**) *p*-Menth-1-en-4-ol, (**11**) Thymol, (**12**) Carvacrol.

	Hepatotoxicity		Carcinogenicity		Cytotoxicity		Immunotoxicity		Mutagenicity		Predicted LD_50_ (mg/kg)	Class
	Pr	Pb	Pr	Pb	Pr	Pb	Pr	Pb	Pr	Pb		
**1**	Inact.	0.86	Inact.	0.55	Inact.	0.98	Inact.	0.78	Inact.	0.73	5000	V
**2**	Inact.	0.86	Inact.	0.60	Inact.	0.99	Inact.	0.93	Inact.	0.75	3700	V
**3**	Inact.	0.81	Inact.	0.59	Inact.	0.51	Inact.	0.82	Inact.	0.71	5000	V
**4**	Inact.	0.80	Inact.	0.66	Inact.	0.97	Inact.	0.95	Inact.	0.71	4700	V
**5**	Inact.	0.77	Inact.	0.60	Inact.	0.99	Inact.	0.98	Inact.	0.75	5000	V
**6**	Inact.	0.86	Inact.	0.56	Inact.	0.95	Inact.	0.72	Inact.	0.76	4800	V
**7**	Inact.	0.87	**Act.**	**0.67**	Inact.	0.98	Inact.	0.98	Inact.	0.89	2374	V
**8**	Inact.	0.76	Inact.	0.65	Inact.	0.95	Inact.	0.97	Inact.	0.82	4400	V
**9**	Inact.	0.83	Inact.	0.60	Inact.	0.98	Inact.	0.92	Inact.	0.82	2500	V
**10**	Inact.	0.80	Inact.	0.72	Inact.	0.99	Inact.	0.83	Inact.	0.88	1016	IV
**11**	Inact.	0.75	Inact.	0.60	Inact.	0.93	Inact.	0.99	Inact.	0.89	640	IV
**12**	Inact.	0.75	Inact.	0.60	Inact.	0.96	Inact.	0.99	Inact.	0.89	810	IV

Pr: Prediction, Pb: Probability, Inact.: Inactive, Act.: Active. Toxicity class explanation: Class IV: substances that are harmful if swallowed (LD_50_ ranging from 300 to 2000 mg/kg), Class V: compounds that may be harmful if swallowed (LD_50_ ranging between 2000 and 5000 mg/kg).

**Table 4 antibiotics-12-00655-t004:** Heat map of the docking scores (affinity values are expressed in kcal/mol) of *P. vericillata* components.

N°	Compounds	Antibacterial Proteins (PDB IDs)				Antifungal Proteins (PDB IDs)		Antioxidant Proteins (PDB IDs)			
		1KZN	3GNS	2VF5	3VSL	1EA1	1IYL	1N8Q	1OG5	2CDU	4JK4
		Free Binding Energy (Kcal/mol) *									
**-**	Native Ligand	−9.6	−6.2	−7.2	−7.8	−5.8	−5.8	−6	−6.6	−8.6	−5.3
**1**	α-Thujene	−5.3	−4.7	−4.9	−4.6	−4.4	−4.5	−6.5	−5.6	−5.7	−5.8
**2**	α-Pinene	−4.7	−5	−5.2	−4.7	−4.2	−4.9	−5.6	−5.6	−5.5	−5.9
**3**	Sabinene	−5.5	−4.7	−4.8	−4.7	−4	−4.6	−5.4	−6.2	−5.4	−5.9
**4**	β-Pinene	−4.7	−4.9	−5.2	−4.7	−4.1	−4.8	−5.5	−5.6	−5.6	−5.9
**5**	β-Myrcene	−4.9	−4.3	−4.3	−4.2	−3.2	−4.6	−4.1	−5.3	−4.8	−6.4
**6**	(+)-4-Carene	−5	−5.1	−5.1	−4.8	−6.1	−4.8	−6.1	−5.8	−5.8	−6.2
**7**	*m*-Cymene	−6	−4.9	−5.1	−4.9	−4.2	−4.8	−4.1	−5.9	−5.8	−7.4
**8**	D-Limonene	−5.8	−4.7	−5	−4.6	−4.4	−4.6	−6	−6.3	−5.6	−6
**9**	γ-Terpinene	−5.8	−4.7	−5	−4.6	−4.3	−4.7	−5.1	−6.1	−5.6	−7
**10**	*p*-Menth-1-en-4-ol	−6.2	−5.2	−5.5	−5.1	−4.4	−4.8	−5.8	−5.9	−5.7	−6.2
**11**	Thymol	−6.2	−5.1	−5.2	−5.1	−4.7	−4.7	−6.2	−6	−5.5	−6.2
**12**	Carvacrol	−6	−5.4	−5.3	−5.1	−4.7	−5.3	−6.2	−6.3	−6	−6.4

* The color scale for each column runs from red (representing the native ligand ∆G), through yellow (the midpoint), to green (representing the native ligand ∆G +4 kcal/mol). 1KZN: DNA Gyrase Topoisomerase II; 3GNS: Enoyl-Acyl Carrier Protein Reductase; 2VF5: Glucosamine-6-Phosphate Synthase; 3VSL: Penicillin Binding Protein 3; 1EA1: Cytochrome P450 14 Alpha-Sterol Demethylase; 1IYL: N-Myristoyl Transferase; 1N8Q: Lipoxygenase; 1OG5: CYP2C9; 2CDU: NADPH Oxidase; 4JK4: Bovine Serum Albumin.

**Table 5 antibiotics-12-00655-t005:** Assessment of PVEO’s antioxidant potential through DPPH and FRAP tests.

Antioxidant Tests	DPPH Assay	FRAP Test
	Inhibitory Concentration 50 (μg/mL)	
PVEO	70.6 ± 0.005	50 ± 0.001
Ascorbic acid *	21.06 ± 0.001	19.33 ± 0.001

* Ascorbic acid, considered as positive control.

**Table 6 antibiotics-12-00655-t006:** Evaluation of the minimum inhibitory concentration and the bactericidal concentration of PVEO.

Bacterial Strains	PVEO				Gentamicin (1 mg/mL)
	MIC (%)	MBC (%)	MBC/MIC	IZ (mm)	IZ (mm)
*L. innocua*	0.5	1	2	23.00	21.50
*E. coli*	0.25	1	4	46.25	22.50
*S. aureus*	0.25	1	4	36.50	19.50
*P. aeruginosa*	0.5	1	2	22.50	20.50

**Table 7 antibiotics-12-00655-t007:** Assessment of the minimum inhibitory and fungicidal concentrations of PVEO against fungal strains.

Fungal Strains	PVEO				Cycloheximide (1 mg/mL)
	MIC (%)	MFC (%)	MFC/MIC	IZ (mm)	IZ (mm)
*R. glutinis*	1	1	1	19.50	21.00
*G. candidum*	1	1	1	31.75	23.00
*C. albicans*	1	1	1	29.50	21.40

**Table 8 antibiotics-12-00655-t008:** Molecular modeling targets and grid box characteristics.

Proteins/PDB IDs	Grid Box Size	Gride Box Center	Native Ligand	Reference
DNA Gyrase Topoisomerase II (*E. coli*)/**1KZN**	size_x = 40 size_y = 40 size_z = 40	center_x = 19.528 center_y = 19.500 center_z = 43.031	Clorobiocin	[37]
Glucosamine-6-Phosphate Synthase/**2VF5**	size_x = 70 size_y = 64 size_z = 56	center_x = 30.590 center_y = 15.822 center_z = 3.4970	Glucosamine-6-Phosphate	[36]
Enoyl-Acyl Carrier Protein Reductase (*S. aureus*)/**3GNS**	size_x = 40 size_y = 40 size_z = 40	center_x = −14.280 center_y = 0.56200 center_z = −21.462	Triclosan	[38,39]
Penicillin Binding Protein 3/**3VSL**	size_x = 40 size_y = 40 size_z = 40	center_x = 18.391 center_y = −48.761 center_z = 23.317	Cefotaxime	[34,35]
Cytochrome P450 14 Alpha-Sterol Demethylase/**1EA1**	size_x = 40 size_y = 40 size_z = 40	center_x = 17.702 center_y = −3.978 center_z = 67.221	Fluconazole	[34,35]
N-Myristoyl Transferase/**1IYL**	size_x = 40 size_y = 40 size_z = 40	center_x = −11.256 center_y = 49.991 center_z = 1.040	Fluconazole	[34,35]
Lipoxygenase/**1N8Q**	size_x = 40 size_y = 40 size_z = 40	center_x = 22.455 center_y = 1.2930 center_z = 20.362	Protocatechuic Acid	[65]
CYP2C9/**1OG5**	size_x = 12.387 size_y = 11.653 size_z = 11.654	center_x = −19.823 center_y = 86.686 center_z = 38.275	Warfarin	[65]
NADPH Oxidase/**2CDU**	size_x = 14.007 size_y = 14.997 size_z = 18.795	center_x = 18.997 center_y = −5.777 center_z = −1.808	Adenosine-5′-Diphosphate	[65,81]

## Data Availability

Not applicable.

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
