# Peer review of "Assessment of the Antioxidant and Antimicrobial Potential of Ptychotis verticillata Duby Essential Oil from Eastern Morocco: An In Vitro and In Silico Analysis"

_antibiotics, 2023, doi:10.3390/antibiotics12040655_

Round 1
Reviewer 1 Report
The manuscript antibiotics-1 2259503 “Assessment of the Antioxidant and Antimicrobial Potential of Ptychotis verticillata Duby Essential Oil from Eastern Morocco: An In Vitro and In Silico Analysis” is devoted to the investigation of essential oil from Ptychotis verticillata Duby – its chemical composition, in vitro and in silico antioxidant and antimicrobial properties.
I am sorry but I have many questions to this work
1. What is the novelty of your work? In the works you cited in the introduction, the results are similar: The chemical composition of the essential oil, antimicrobial and antioxidant activity were studied before.
Bnowham, M.; Benalla, W.; Asehraou, A.; Berrabah, M. Antibacterial Activity of Essential Oil from Ptychotis Verticillata. 661 Spat. D.D. 2012, 2, 69–73. 662 16. Tomi, P.; Bouyanzer, A.; Hammouti, B.; Desjobert, J.-M.; Costa, J.; Paolini, J. Chemical Composition and Antioxidant 663 Activity of Essential Oils and Solvent Extracts of Ptychotis Verticillata from Morocco. food chem. Toxicol. 2011, 49, 533–536.
They also describe the chemical composition of the essential oil, as well as the results of the study of antimicrobial and antioxidant activity. Therefore, there is no scientific novelty in the work
2. It is not clear how reliable your results are, especially since you write that the plant has practically not been studied. Did you study just one sample? What could be the difference in essential oil yield? In table. 2 need to add the results of statistical processing? What was used as a standard in the analysis?
3. In my opinion, your research in silico is not logical. It is unlikely that the essential oil will be used internally, so it is illogical to study the pharmacokinetic parameters for it (especially for each component of the oil separately). Essential oil is the sum of biologically active substances. Their number is quite large and how to take into account their contribution, taking into account the rather different content.
4. Docking studies of the components of various essential oils are quite well described in the literature, for example, among many:
Aouf, A.; Bouaouina, S.; Abdelgawad, M.A.; Abourehab, M.A.S.; Farouk, A. In Silico Study for Algerian Essential Oils as Antimicrobial Agents against Multidrug-Resistant Bacteria Isolated from Pus Samples. Antibiotics 2022, 11, 1317. https://doi.org/10.3390/antibiotics11101317
Kandsi, F.; Elbouzidi, A.; Lafdil, F.Z.; Meskali, N.; Azhar, A.; Addi, M.; Hano, C.; Maleb, A.; Gseyra, N. Antibacterial and Antioxidant Activity of Dysphania ambrosioides (L.) Mosyakin and Clemants Essential Oils: Experimental and Computational Approaches. Antibiotics 2022, 11, 482. https://doi.org/10.3390/antibiotics11040482
These results will be repeated as current programs will always give the same results for the same chemicals.
5 The in vitro methods used for determining antioxidant activity are classified as antiradical and it is incorrect and pointless to calculate affinity for receptors for them.
My decision is reject
Author Response
Dear Editors, Dear Reviewers,
Thank you for giving us the opportunity to improve our manuscript with the revised version and thank you for your useful comments.
We really thank the Reviewers for their thorough review. We hope this revision will satisfy reviewers’ queries, and that our work will be considered for publication in Antibiotics. Below we have made efforts to either address or respond to each (paraphrased) requested change and communicated weakness. Major changes are highlighted in yellow text in the revision. All typos/minor concerns have been fixed in the manuscript and are otherwise not addressed in this response.
With kind regards
Dr. Hano, Dr. Addi, and the co-Authors
Reviewer 1. Query 1. What is the novelty of your work? In the works you cited in the introduction, the results are similar: The chemical composition of the essential oil, antimicrobial and antioxidant activity were studied before.
Bnowham, M.; Benalla, W.; Asehraou, A.; Berrabah, M. Antibacterial Activity of Essential Oil from Ptychotis Verticillata. 661 Spat. D.D. 2012, 2, 69–73. 662 16.
Tomi, P.; Bouyanzer, A.; Hammouti, B.; Desjobert, J.-M.; Costa, J.; Paolini, J. Chemical Composition and Antioxidant 663 Activity of Essential Oils and Solvent Extracts of Ptychotis Verticillata from Morocco. food chem. Toxicol. 2011, 49, 533–536.
They also describe the chemical composition of the essential oil, as well as the results of the study of antimicrobial and antioxidant activity. Therefore, there is no scientific novelty in the work
Response. Thank you for your time used to give the insightful and vigilant comments. In the paper untitled “Chemical composition and antioxidant activity of essential oils and solvent extracts of Ptychotis verticillata from Morocco”, the authors investigated only the antioxidant activity of the PVEO using only DPPH test system. It is important to emphasize that antioxidant activity is a complex procedure usually happening through several mechanisms and is influenced by many factors, which cannot be fully described with one single method. Therefore, it is essential to perform more than one type of antioxidant capacity measurement to take into account the various mechanisms. Thus, in our study, the antioxidant potential of PVEO was evaluated and confirmed using two methods with two different mechanisms: the 2,2-diphenyl-1-picrylhydrazyl (DPPH) radical trapping assay and the ferric reducing antioxidant power (FRAP) method. The DPPH method is based on electron or hydrogen transfer to the stable DPPH free radical. While FRAP method is based on the reduction of ferric (Fe3+) to ferrous (Fe2+) ion through the donation of an electron.
In the second paper entitled “Antibacterial activity of essential oil from Ptychotis verticillata", the authors investigated the antibacterial activity of PVEO using only agar diffusion method. Yet, agar diffusion method provides qualitative results by showing bacterial growth inhibition. This latter does not mean bacterial death and consequently, this method cannot distinguish bactericidal and bacteriostatic effects. Moreover, agar diffusion method is not appropriate to determine the minimum inhibitory concentration (MIC) as it is impossible to quantify the amount of the antimicrobial agent diffused into the agar medium. Our study is the first one that investigates the antimicrobial potential of PVEO quantitatively by the determination of its minimum inhibitory concentration (MIC), minimum bactericidal concentration (MBC), and minimum fungicidal concentration (MFC).
Reviewer 1. Query 2. It is not clear how reliable your results are, especially since you write that the plant has practically not been studied. Did you study just one sample? What could be the difference in essential oil yield? In table. 2 need to add the results of statistical processing? What was used as a standard in the analysis?
Response. To begin, I want to express my gratitude for your observation. I'd like to inform the reviewer that we conducted gaz chromatography analysis using a Shimadzu GC-2010 gaz chromatography coupled with a mass spectrometer detector (GC-MS-QP2010), thus no standards are needed to confirm the chemical composition. In fact, the results depicted on the figure and the table represent the qualitative and semi-quantitative analysis of PVEO. Essential oil constituents were identified by comparing their MS data with the NIST147 computer library. Data collection and processing were performed using LabSolutions (version 2.5). The relative percentage of each component of the Eos under investigation was detected based on their GC peak areas. It's worth noting that this equipment provides highly accurate qualitative and semi-quantitative information on the chemical composition of essential oils thus no repetition is needed to confirm the performance of the machine. Furthermore, I'd like to point out that the studied plant is endemic to Morocco, indicating its availability in the local market where it was purchased. I totally understand your concerns about the essential oil yield, but I would like to note that the quantity of essential oil produced is greatly impacted by various factors including the time of harvesting, extraction method, climate and soil conditions, plant age, and other environmental factors.
Reviewer 1. Query 3. In my opinion, your research in silico is not logical. It is unlikely that the essential oil will be used internally, so it is illogical to study the pharmacokinetic parameters for it (especially for each component of the oil separately). Essential oil is the sum of biologically active substances. Their number is quite large and how to take into account their contribution, taking into account the rather different content.
Response. Thank you for your valuable comment, It is understandable that you may have concerns about the practicality of studying the pharmacokinetic parameters of each component of an essential oil in silico. However, it is important to note that in silico studies can provide valuable insights into the potential bioactivity and toxicity of individual components within the essential oil. Furthermore, while it may be unlikely that the essential oil will be used internally, it is still possible that it may be applied topically or used in aromatherapy, or even as a food preservative; A study conducted by Burt (2004) [1], deals with the antimicrobial activity of a combination of essential oils (including peppermint, cinnamon, and oregano) against certain foodborne pathogens, and found that the oils were effective in inhibiting the growth of the bacteria when added to food. Although certain essential oils are categorized as food-grade and may be utilized in limited quantities as flavoring agents or dietary supplements, there are research findings that indicate the potential oral usage of essential oils. For instance, Barocelli et al. (2004) [2] conducted a study that proposes the use of Lavandula hybrida Reverchon “Grosso” essential oil, either via inhalation or oral administration, due to its antinociceptive and gastroprotective properties, among other relevant studies from the literature. In such cases, understanding the pharmacokinetics of the individual components can help inform the safe and effective use of the essential oil in various applications.
Reviewer 1. Query 4. Docking studies of the components of various essential oils are quite well described in the literature, for example, among many:
Aouf, A.; Bouaouina, S.; Abdelgawad, M.A.; Abourehab, M.A.S.; Farouk, A. In Silico Study for Algerian Essential Oils as Antimicrobial Agents against Multidrug-Resistant Bacteria Isolated from Pus Samples. Antibiotics 2022, 11, 1317. https://doi.org/10.3390/antibiotics11101317
Kandsi, F.; Elbouzidi, A.; Lafdil, F.Z.; Meskali, N.; Azhar, A.; Addi, M.; Hano, C.; Maleb, A.; Gseyra, N. Antibacterial and Antioxidant Activity of Dysphania ambrosioides (L.) Mosyakin and Clemants Essential Oils: Experimental and Computational Approaches. Antibiotics 2022, 11, 482. https://doi.org/10.3390/antibiotics11040482
These results will be repeated as current programs will always give the same results for the same chemicals.
Response. While it is true that docking studies using the same programs will always produce the same results for the same compounds, it's important to note that the protocols used to conduct these studies can differ significantly from one study to another. As it is evident that not all the identified compounds in PVEO were studied in complex with the targeted antibacterial, antifungal, and antioxidant proteins, at once, our study evaluates the chemical makeup of PVEO as a whole to reveal which compound (among the identified ones), is the one that may be responsible of the observed experimental findings, regarding the antibacterial, antifungal, and the antioxidant results.
Reviewer 1. Query 5. The in vitro methods used for determining antioxidant activity are classified as antiradical and it is incorrect and pointless to calculate affinity for receptors for them.
Response. We thank the reviewer for his remark. While it is true that in vitro methods used for determining antioxidant activity are typically classified as antiradical and may not involve direct binding to specific receptors, working on antioxidant proteins in silico can still be necessary and informative [3–5]. Furthermore, while in vitro antiradical methods are useful for evaluating the ability of a compound to scavenge free radicals and prevent oxidative damage, they do not necessarily reflect the complex interactions and dynamic processes that occur within living cells and tissues. In silico studies of antioxidant proteins can help to bridge this gap by providing a more comprehensive understanding of the underlying mechanisms of the antioxidant activity. Therefore, while it may not be necessary to calculate affinity for receptors when assessing antioxidant activity using in silico molecular docking using common antioxidant proteins (such as lipoxygenases, NADPH-oxidase …), It can still be necessary and valuable for understanding the broader context of antioxidant defense mechanisms and developing new antioxidant therapies. The chosen antioxidant proteins were retrieved from the relevant literature.
References:
- Burt, S. Essential Oils: Their Antibacterial Properties and Potential Applications in Foods--a Review. Int. J. Food Microbiol. 2004, 94, 223–253, doi:10.1016/j.ijfoodmicro.2004.03.022.
- Barocelli, E.; Calcina, F.; Chiavarini, M.; Impicciatore, M.; Bruni, R.; Bianchi, A.; Ballabeni, V. Antinociceptive and Gastroprotective Effects of Inhaled and Orally Administered Lavandula Hybrida Reverchon “Grosso” Essential Oil. Life Sci. 2004, 76, 213–223.
- Jianu, C.; Stoin, D.; Cocan, I.; David, I.; Pop, G.; Lukinich-Gruia, A.T.; Mioc, M.; Mioc, A.; Șoica, C.; Muntean, D. In Silico and in Vitro Evaluation of the Antimicrobial and Antioxidant Potential of Mentha× Smithiana R. GRAHAM Essential Oil from Western Romania. Foods 2021, 10, 815.
- Rădulescu, M.; Jianu, C.; Lukinich-Gruia, A.T.; Mioc, M.; Mioc, A.; Șoica, C.; Stana, L.G. Chemical Composition, in Vitro and in Silico Antioxidant Potential of Melissa Officinalis Subsp. Officinalis Essential Oil. Antioxidants 2021, 10, 1081.
- Herrera-Calderon, O.; Chacaltana-Ramos, L.J.; Huayanca-Gutiérrez, I.C.; Algarni, M.A.; Alqarni, M.; Batiha, G.E.-S. Chemical Constituents, In Vitro Antioxidant Activity and In Silico Study on NADPH Oxidase of Allium Sativum L.(Garlic) Essential Oil. Antioxidants 2021, 10, 1844.

Reviewer 2 Report
The authors described the “Assessment of the Antioxidant and Antimicrobial Potential of Ptychotis verticillata Duby Essential Oil from Eastern Morocco: An In Vitro and In Silico Analysis”. The goal of this research is to uncover the phytochemical makeup of the essential oil extracted from P. verticillata, and evaluate their Antioxidant and Antimicrobial activities. I consider that the manuscript meets the requirements to be published after major revision. Additional suggestions and comments are included:
1. GC-MS Chromatogram of the Chemical Composition of P. Verticillata Essential Oil PVEO is provided in the manuscript. Copies of MS of chemical composition also need to be provided.
2. As described by the authors“The antioxidant capacity observed in our analyzed EO is most likely attributed to its chemical makeup, with phenolic compounds being the dominant component. Phenolic compounds are considered highly effective natural antioxidants due to their chemical structure, which enables them to scavenge free radicals and convert them into stable compounds through proton and electron transfer mechanisms.”
The content of phenolic compounds in different parts of the plant is different, and will also change with the season, which will directly affect the antioxidant activity of the extract. What parts of the plant are used by the author? In what season?
4. Almost all 12 kinds of PVEO’s compounds can be purchased directly. It is suggested that the author can directly use pure substances for comparative experiments to verify the accuracy of the prediction method.
5. The formula of the scavenging activity is very blurry.
Author Response
Dear Editors, Dear Reviewers,
Thank you for giving us the opportunity to improve our manuscript with the revised version and thank you for your useful comments.
We really thank the Reviewers for their thorough review. We hope this revision will satisfy reviewers’ queries, and that our work will be considered for publication in Antibiotics. Below we have made efforts to either address or respond to each (paraphrased) requested change and communicated weakness. Major changes are highlighted in yellow text in the revision. All typos/minor concerns have been fixed in the manuscript and are otherwise not addressed in this response.
With kind regards
Dr. Hano, Dr. Addi, and the co-Authors
Reviewer 2. Comment. The authors described the “Assessment of the Antioxidant and Antimicrobial Potential of Ptychotis verticillata Duby Essential Oil from Eastern Morocco: An In Vitro and In Silico Analysis”. The goal of this research is to uncover the phytochemical makeup of the essential oil extracted from P. verticillata, and evaluate their Antioxidant and Antimicrobial activities. I consider that the manuscript meets the requirements to be published after major revision.
Response. We would like to express our appreciation to Reviewer 2, for his/her suggestions, as we believe that helped us improving the quality of our paper.
Additional suggestions and comments are included:
Reviewer 2. Query 1. GC-MS Chromatogram of the Chemical Composition of P. Verticillata Essential Oil (PVEO) is provided in the manuscript. Copies of MS of chemical composition also need to be provided.
Response. Thank you for your time used to give the insightful and vigilant comments. copies of the MS of the chemical composition were added to our supplementary file.
Reviewer 2. Query 2. As described by the authors “The antioxidant capacity observed in our analyzed EO is most likely attributed to its chemical makeup, with phenolic compounds being the dominant component. Phenolic compounds are considered highly effective natural antioxidants due to their chemical structure, which enables them to scavenge free radicals and convert them into stable compounds through proton and electron transfer mechanisms.”
The content of phenolic compounds in different parts of the plant is different, and will also change with the season, which will directly affect the antioxidant activity of the extract. What parts of the plant are used by the author? In what season?
Response. Thank you for your sharp remark. The plant P. verticillata used in this study was harvested during the spring season and the aerial part was used for the extraction of the essential oil (this information was added to the manuscript, and highlighted in yellow)
Reviewer 2. Query 3. Almost all 12 kinds of PVEO’s compounds can be purchased directly. It is suggested that the author can directly use pure substances for comparative experiments to verify the accuracy of the prediction method.
Response. The authors would like to thank you for your valuable and highly constructive suggestion. The main purpose of our study is to valorize the studied plant, and particularly its essential oil. Currently, the compounds are not available in our laboratory. However, we will take your valuable suggestion into account for our future studies.
Reviewer 2. Query 4. The formula of the scavenging activity is very blurry.
Response. Thank you, the formula has been rewritten. You will find it highlighted in yellow in our main manuscript.

Reviewer 3 Report
Assessment of the Antioxidant and Antimicrobial Potential of 2 Ptychotis verticillata Duby Essential Oil from Eastern Moroc- 3 co: An In Vitro and In Silico Analysis deals with uncovering the phytochemical makeup of the essential oil extracted from P. verticillata, which is indigenous to the Touissite region in Eastern Morocco.
Authors must add more potential lines about potential metabolites obtained in their study correlating with other recent data
“Additionally, none of the phytoconstituents were identified as substrates for Renal 164 OCT2 (Organic Cation Transporter 2), with β-Myrcene exhibiting the best total clearance (in 165 mL/min/kg).”
Explain this observation
Don’t write “using a heat-map style table,”, mention it as heat map table
Authors must explain in more detail, “3.4. PASS, ADME, the prediction of the Toxicity Analysis (Pro-Tox II)”
Please add petri plate figure as indicator of antimicrobial assay
In introduction, following papers may be cited for importance of medicinal plants
Evaluation of aflatoxin contamination in crude medicinal plants used for the preparation of herbal medicine
Oriental Pharmacy and Experimental Medicine 19 (2), 137-143
Antioxidant and antimicrobial activity displayed by a fungal endophyte Alternaria alternata isolated from Picrorhiza kurroa from Garhwal Himalayas, India
Biocatalysis and Agricultural Biotechnology, 101955
Somewhere grammar is minor problem. Overall a satisfactory paper and author must edit as per above
Author Response
Dear Editors, Dear Reviewers,
Thank you for giving us the opportunity to improve our manuscript with the revised version and thank you for your useful comments.
We really thank the Reviewers for their thorough review. We hope this revision will satisfy reviewers’ queries, and that our work will be considered for publication in Antibiotics. Below we have made efforts to either address or respond to each (paraphrased) requested change and communicated weakness. Major changes are highlighted in yellow text in the revision. All typos/minor concerns have been fixed in the manuscript and are otherwise not addressed in this response.
With kind regards
Dr. Hano, Dr. Addi, and the co-Authors
Reviewer 3. Query 1. “Additionally, none of the phytoconstituents were identified as substrates for Renal OCT2 (Organic Cation Transporter 2), with β-Myrcene exhibiting the best total clearance (in 165 mL/min/kg).”. Explain this observation.
Response. Organic cations are positively charged molecules that are important in drug transport and elimination. Renal OCT2 is responsible for transporting many organic cations, including some drugs, out of the body through the urine. Testing whether the PVEO’s identified molecules are OCT2 substrate provides insights into how these molecules are processed and eliminated by the body, which could have implications for understanding the potential health benefits and risks of consuming these compounds. We have added expanded further this observation in our manuscript (the added text is as below). Thank you for your valuable remark.
“In this study, we have tested in silico the ability of the various phytoconstituents to be transported by Renal OCT2. Organic cations are positively charged molecules that are important in drug transport and elimination. Renal OCT2 is responsible for transporting many organic cations, including some drugs, out of the body through the urine. Surprisingly, none of the phytoconstituents were identified as substrates for this transporter, meaning they were not effectively transported by Renal OCT2. The phytoconstituent β-Myrcene had the best total clearance, with a rate of 0.438 mL/min/kg. This suggests that β-Myrcene is efficiently eliminated from the body through processes other than Renal OCT2 transport.”
Reviewer 3. Query 2. Don’t write “using a heat-map style table,”, mention it as heat map table.
Response. Thank you for your suggestion. We have rectified it in the main text.
Reviewer 3. Query 3. Authors must explain in more detail, “3.4. PASS, ADME, the prediction of the Toxicity Analysis (Pro-Tox II)”
Response. We have added more details to this section (section 3.4.), thank you for your comment.
Reviewer 3. Query 4. Please add petri plate figure as indicator of antimicrobial assay.
Response. Thank you for your suggestion. We have added petri plate figure to our manuscript.
Reviewer 3. Query 5. In introduction, following papers may be cited for importance of medicinal plants.
Evaluation of aflatoxin contamination in crude medicinal plants used for the preparation of herbal medicine. Oriental Pharmacy and Experimental Medicine. 19 (2), 137-143
Antioxidant and antimicrobial activity displayed by a fungal endophyte Alternaria alternata isolated from Picrorhiza kurroa from Garhwal Himalayas, India. Biocatalysis and Agricultural Biotechnology, 101955
Response. Thank you for the suggested papers. The two papers have been cited in the introduction part as suggested as they are relevant to our present work.

Round 2
Reviewer 1 Report
Dear authors, I have read your response and explanation. But unfortunatelly this doesn't change my opinion about your manuscript. Your experiment is good but not novel? and design of your investigation is not appropriate. I'd recommend your reorganise it completely and give all information in comparison with another samples from different parts of country or maybe from different countries.
I'm sorry but in presented variant I can't reccommend your manuscript to be published.
Author Response
Dear Esteemed Reviewer,
Please allow me to express my sincere gratitude for the invaluable time and effort that you have dedicated to evaluating our manuscript. Your constructive feedback has been immensely helpful in improving the quality and rigor of our work.
However, I must humbly express my regret that our justifications regarding the novelty and originality of our study did not fully convince you. We have taken great care to reinforce these arguments with indexed articles, as can be seen in the revised manuscript.
I would like to draw your attention to the fact that our study focuses on Ptychotis verticillata Duby., which is an endemic plant of the eastern region, exclusive to this particular area. Therefore, it is not appropriate to compare our results with other regions of the country or the world. Furthermore, a composition comparison has already been carried out with another study that focused on the same plant. With regards to the antimicrobial activity of this
plant, our study is unique as we are the only ones to have conducted quantitative tests (MIC, MBC, and MFC). It would be illogical to compare these results with other qualitative tests (zones of inhibition).
Finally, our findings scientifically confirm the ethnomedicinal usage and potential
therapeutic benefits of this plant. It may serve as a promising source for future
pharmaceutical development.
Once again, thank you for your valuable feedback and please do not hesitate to
contact us if you require any further information.
Sincerely,
Dr. Addi, Dr. Hano, and the Co-Authors
Reviewer 2 Report
The authors have made careful modifications according to the review comments, and it is recommended to accept it directly.
Author Response
We would like to take this opportunity to convey our sincere appreciation to the
esteemed Reviewer 2 for his/her valuable input, which has undoubtedly played a crucial role in enhancing the overall quality of our manuscript. The insightful comments and suggestions provided by Reviewer 2 have been of immense help in refining and strengthening our research work, and we are truly grateful for his/her efforts.